**Subject Category:**
Biology (whole organism)

behaviour/ecology

flipper, foraging, behavioural plasticity, aerial footage, Southeast Alaska, humpback whale

**Author for correspondence:**
Madison M. Kosma
e-mail: madison.kosma@gmail.com

# Pectoral herding: an innovative tactic for humpback whale foraging

Madison M. Kosma[1], Alexander J. Werth[2], Andrew R. Szabo[3] and Janice M. Straley[4]

[1]College of Fisheries and Ocean Sciences, University of Alaska Fairbanks, Juneau, AK 99801, USA
[2]Department of Biology, Hampden-Sydney College, Hampden-Sydney, VA 23943, USA
[3]Alaska Whale Foundation, Petersburg, AK 99833, USA
[4]Department of Natural Sciences, University of Alaska Southeast, Sitka, AK 99835, USA

 MMK, 0000-0002-7323-4487; AJW, 0000-0002-7777-478X

Humpback whales (*Megaptera novaeangliae*) have exceptionally long pectorals (i.e. flippers) that aid in shallow water navigation, rapid acceleration and increased manoeuvrability. The use of pectorals to herd or manipulate prey has been hypothesized since the 1930s. We combined new technology and a unique viewing platform to document the additional use of pectorals to aggregate prey during foraging events. Here, we provide a description of 'pectoral herding' and explore the conditions that may promote this innovative foraging behaviour. Specifically, we analysed aerial videos and photographic sequences to assess the function of pectorals during feeding events near salmon hatchery release sites in Southeast Alaska (2016–2018). We observed the use of solo bubble-nets to initially corral prey, followed by calculated movements to establish a secondary boundary with the pectorals—further condensing prey and increasing foraging efficiency. We found three ways in which humpback whales use pectorals to herd prey: (i) create a physical barrier to prevent evasion, (ii) cause water motion to guide prey towards the mouth, and (iii) position the ventral side to reflect light and alter prey movement. Our findings suggest that behavioural plasticity may aid foraging in changing environments and shifts in prey availability. Further study would clarify if 'pectoral herding' is used as a principal foraging tool by the broader humpback whale population and the conditions that promote its use.

# 1. Background

Large body sizes of baleen whales generate high metabolic demands that require the consumption of sizable, dense patches

of prey [1–3]. However, filter feeding is energetically demanding and requires effective methods for prey aggregation [2]. Behavioural plasticity and foraging innovations are common among rorquals [4,5]. Humpback whales (*Megaptera novaeangliae*) provide an excellent example of how individual changes in behaviour can lead to diverse foraging tactics that maximize feeding efficiency [6–9]. Such foraging includes lunge feeding [6,10], bubble-net feeding [6,11–14], flick feeding [6], cooperative feeding [15], lobtail feeding [7] and other idiosyncratic tactics [12,16–18].

Humpback whales are one of the world's largest filter-feeders and regularly use lunge feeding to capture prey. This particular technique is energetically costly [19] and requires a two-step process. The whale first uses a high-velocity lunge to engulf large volumes of prey-laden water. The whale then closes its mouth and the baleen acts as a sieve to filter prey [14,20]. The lunge can occur at depth [2,10,20–22] or on the surface [7,23,24]. In both situations, lunge feeding requires acceleration to high speeds [2,25] because the animal must overcome considerable drag from an open mouth. To counteract drag and increase speed, humpback whales open their mouths gradually, in synchrony with strong fluke strokes [20,22]. This acceleration maximizes the amount of water engulfed and aids in the capture of active prey [25]. Humpback whales feeding near the surface exhibit an array of lunge types [6,12,15] and some are in association with the creation of bubbles. A bubble-net is denoted by the formation of a ring of bubbles in a clockwise fashion to enclose prey [6,7,12,13,26] and this strategy can be employed by an individual or a group of whales. Bubble-nets serve as a physical barrier to increase lunge efficiencies and are most commonly used on naturally schooling fish (i.e. Pacific herring).

Humpback whales have a distinctive body morphology that allows for the efficient capture of prey [27,28]. Notably, they have the longest pectorals (i.e. flippers) of any cetacean, measuring from one-quarter to one-third of their body length [29,30]. The pectorals of other cetaceans typically do not exceed one-seventh the length of their bodies [31]. The exceptionally long appendages of humpback whales allow for effective navigation in shallower water [31,32], rapid acceleration, greater manoeuvrability and increased stability [6,33,34], thereby increasing capture abilities of small prey such as euphausiids, herring (*Clupea* spp.), capelin (*Mallotus villosus*) and sandlance (*Ammodytes* spp.) [31,35–37]. If not positioned effectively, however, larger pectorals may present a hydrodynamic disadvantage by increasing drag [38].

As the buccal cavity expands during a lunge, a hydrodynamically optimal position for the pectorals is for one or both to extend with the leading edge held at low angles of attack ($\alpha$) [39]. Positioning the pectorals in this manner minimizes drag and provides the greatest amount of lift. The perpendicular position of extended pectorals also stabilizes the whale's body during a lunge [39]. Additionally, it has been hypothesized that rapid pectoral movement just prior to a lunge generates an upward pitching motion that counteracts the torque caused by rapidly engulfing water [34,39]. Segre *et al.* [40] defined four conditions for pectoral movement that would generate lift and increase propulsive thrust during an engulfment event: (i) both pectorals must move symmetrically, (ii) pectorals are angled into the path of the stroke, (iii) the stroke is oriented perpendicular to the whale's body, and (iv) the stroke is aligned with the direction of travel [40]. Lift is generated as pectorals are rotated at an angle to the water flow (angle of attack or $\alpha$). However, this angle must be small relative to the direction of travel [41]. Above a critical $\alpha$, the pectoral will impede lift, making the movement detrimental to acceleration. Miklosovic *et al.* [42] found that peak hydrodynamic efficiency of a humpback whale pectoral is around $\alpha = 7.5°$. Above this, drag increases and lift decreases, with complete stall occurring at $\alpha \sim 17.5°$. These studies illustrate that there are strict hydrodynamic criteria for using pectorals efficiently during lunge feeding.

In addition to providing lift, decreasing drag and promoting acceleration, pectorals may be used to corral or concentrate prey during lunge-feeding events. Humpback whales have multiple foraging strategies to aggregate prey, but concentration of prey may be increased by herding techniques [31,43]. Howell [43] was the first to suggest that humpback whales use their pectorals to direct schools of fish into their mouths. Brodie [38] elaborated on this theory by describing the use of white coloration on the pectoral's ventral surface to 'flash' fish and herd prey towards the whale's mouth. He stated, 'if there are hydrodynamic disadvantages to such large flippers there must be selective compensation, one possibility being their role in concentrating prey' [38]. Both authors, however, reported reservations about their findings because they lacked the perspective necessary to document such behaviours [38]. Our objective was to use new technology (e.g. unoccupied aerial vehicles (UAVs), small video cameras) to document and describe the distinctive role of humpback whale pectorals in herding and aggregating prey. We focused our efforts on whales feeding near salmon hatchery release sites [44] in Southeast Alaska (2016–2018). Hatchery structures allowed for close approaches with

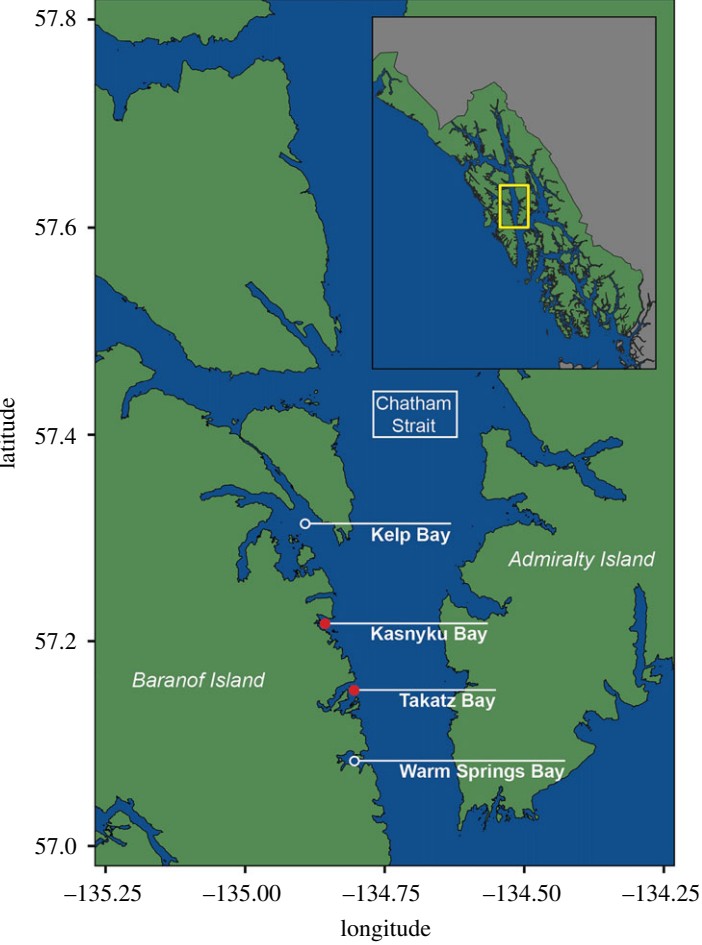

**Figure 1.** Study sites used to document foraging behaviours of humpback whales in Southeast Alaska (2016–2018). Red dots indicate release sites for juvenile hatchery-reared salmon.

minimal behavioural disruption. Our results enhance our understanding of the complex and innovative foraging tactics that may be critical to humpback whale survival as population dynamics and environmental conditions continue to change [45,46].

# 2. Methods

## 2.1. Study location and timing

This study was conducted in Chatham Strait, along the eastern shore of Baranof Island in Southeast Alaska (figure 1). We conducted systematic surveys from Warm Springs Bay north to Kelp Bay, with an emphasis on salmon hatchery release sites in Takatz Bay and Kasnyku Bay in 2016 (mid-May to the end of June) and 2017 (mid-April to the end of July). We put forth a more directed effort to document foraging strategies by humpback whales in Kasnyku Bay in 2018 (May). All effort was timed to overlap with releases of juvenile salmon from Hidden Falls Hatchery (managed by the Northern Southeast Regional Aquaculture Association).

## 2.2. Data collection

We recorded humpback whale sightings and behavioural observations as part of a 3-year study (2016–2018) of humpback whale predation at Hidden Falls Hatchery and surrounding areas. We took identification photographs of each whale using digital SLR cameras with lenses ranging in focal lengths from 70 to 300 mm. Humpback whales were individually identified based on the pigmentation and trailing edges of their flukes and/or the shape and marks of their dorsal fins [47]

and cross-referenced with the Southeast Alaska Humpback Whale Catalog [48]. This catalogue included all whale sightings through 2012 and additional observations from later time periods (JM Straley & CM Gabriele 2016, unpublished data). We made an effort to capture video and photographic sequences with a Nikon D7000 camera whenever whales were observed feeding at the surface. In 2017, we also used a GoPro Hero5 Black video camera affixed to the end of a 3.5 m pole to provide an aerial perspective while standing on walkway platforms attached to hatchery net pens. These platforms provided a unique and close-up perspective without disturbing whale behaviour that enabled camera views directly above or within bubble-nets created by the feeding whales. In 2018, we used an UAV (DJI Mavic Pro with 4 k video at 24 fps) to capture footage of whales surface lunge feeding near the facility. In addition to visual prey identification, we used a cast net and herring jig to sample prey in foraging areas. We removed juvenile salmon otoliths to differentiate hatchery-reared and wild origin fish according to methods described by Volk *et al.* [49].

## 2.3. Data analysis

We used Adobe Premiere Pro to analyse video footage and Adobe Lightroom to assess photographic sequences. Kinematic assessments of whale foraging behaviour were made, with particular focus on the use of pectorals. We recorded pectoral positions, movements and prey locations (when possible) using real-time and frame-by-frame processing. Whale foraging movements were then three-dimensionally modelled using Blender, with post-processing in Adobe Photoshop to accurately illustrate foraging behaviours seen in footage and photographs. Lunge durations were calculated from videos, when possible. All footage and photographic sequences were viewed and categorized based on surface foraging behaviour. Bubble-net feeding was denoted by the formation of a ring of bubbles followed by a lunge through the centre. A surface lunge was recorded as one of two commonly observed types: a vertical lunge, when the animal lunged upwards [24], and a lateral lunge, when the animal rotated approximately 90° while lunging [24]. Pectoral herding, a newly documented feeding strategy, was defined by directed movements of the pectorals to condense prey before a lunge. We identified three ways in which humpback whales used pectorals to herd prey: (i) create a physical barrier to prevent evasion by prey, (ii) cause water motion to direct prey movement, and (iii) position the white coloration on the ventral side to reflect light, causing prey to move in the opposite direction [12,38]. A feeding event was defined as beginning with that start of a solo bubble-net and ending when the whale closed its mouth after a surface lunge. Multiple feeding events from one whale on the same prey, in the same general location, were defined as a foraging session. We calculated lunge duration when possible.

# 3. Results

We captured videos and photographic sequences of two humpback whales independently engaged in previously undocumented foraging techniques. Both whales (Whale A and Whale B) initiated feeding events with a solo bubble-net. Before lunging, these whales used their pectorals to manipulate and further condense prey. We defined this technique as 'pectoral herding', with two methods of execution: 'horizontal pectoral herding' and 'vertical pectoral herding'. More detailed information of Whale A and Whale B encounters are provided in electronic supplementary material, S1 and S2. We captured footage of one additional whale using horizontal pectoral herding, though a limited number of observations precluded this whale from further analyses.

## 3.1. Horizontal pectoral herding

We encountered Whale A (#2360 in Southeast Alaska Humpback Whale Catalog) on 27 days from 2016 to 2018. We observed solo bubble-netting during 15 feeding sessions (135 feeding events). Each solo bubble-net involved what we describe as horizontal pectoral herding prior to the lunge. Video footage depicting horizontal pectoral herding is provided in electronic supplemental material, S3. During horizontal pectoral herding, Whale A initiated the feeding event by deploying an upward-spiral bubble-net to corral prey (figures 2 and 3; Stage A). At the closure of the bubble-net, Whale A rotated its head parallel to the surface of the water and towards the centre of the net. The whale then moved its left pectoral in and out of the water in a forward, sinusoidal motion along the initial edge of the bubble-net barrier (figures 2 and 3; Stage B). Whale A continued this pectoral movement while gradually

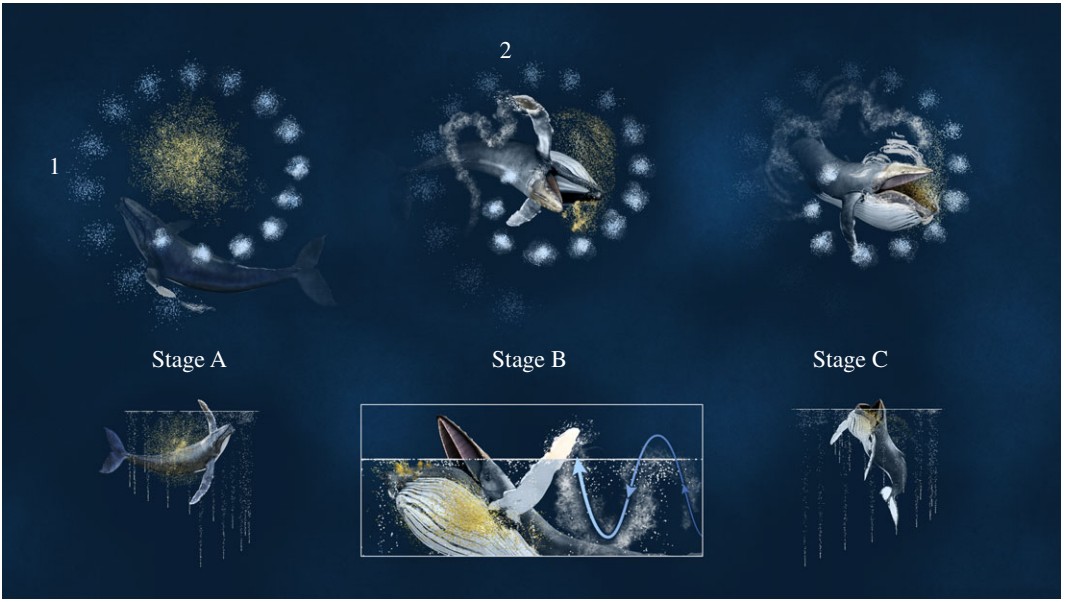

**Figure 2.** Graphical representations of horizontal pectoral herding by Whale A in Southeast Alaska. Prey are denoted in yellow. Stage A: Deployment of an upward-spiral bubble-net to corral the prey and establish the first barrier (1). Stage B: Movement of the left pectoral in and out of the water, along the edge of the bubble-net barrier, creating a secondary barrier (2). Stage C: Lunge to engulf the prey. Graphic by Kyle Kosma.

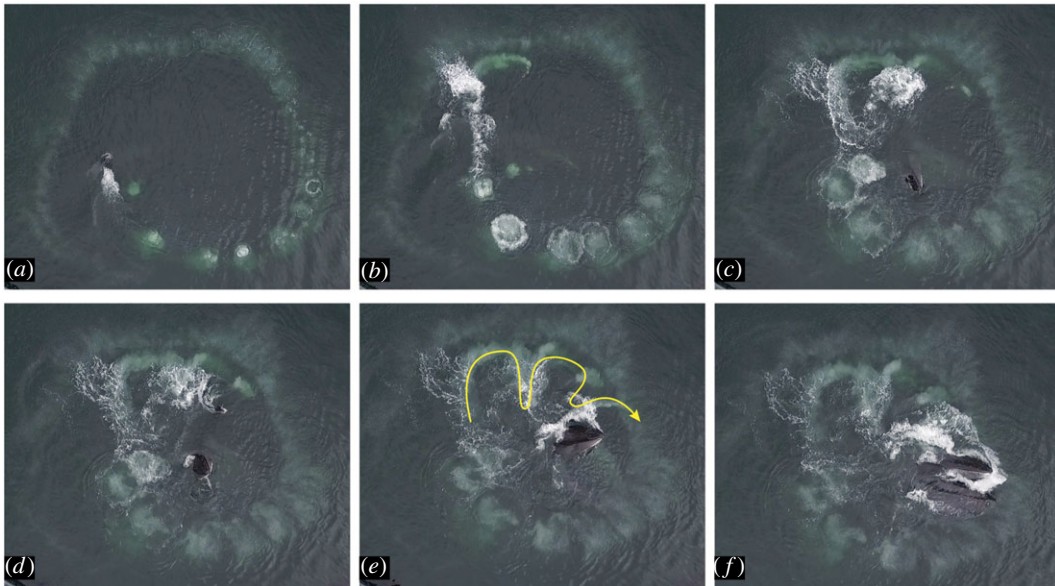

**Figure 3.** Photographic sequence involving horizontal pectoral herding by Whale A in Southeast Alaska. Movements progress from (*a*) beginning to (*f*) end. (*a*) Bubble-net formation; (*b*–*e*), horizontal pectoral herding; (*f*) terminal lunge. Yellow arrow represents the sinusoidal pectoral movement along the edge of the bubble-net barrier.

opening its mouth and allowing the upper jaw to rise above the water line, while the lower jaw remained subsurface. The whale continued to open its mouth wider until it reached the opposite side of the bubble-net (figures 2 and 3; Stage C). Whale A's head rotated in the direction of the left pectoral 51.9% of all documented feeding events. In these cases, the lower jaw was tilted at an angle that exposed prey to the largest circumference of the buccal cavity (figure 4). For all other feeding events, the degree of head tilt was unknown or Whale A maintained a stationary head position, bringing its lower jaw up out of the water to meet the upper jaw. Whale A never rotated its head away from the herding pectoral. The mean lunge duration, defined as the start of pectoral movement to the close of the

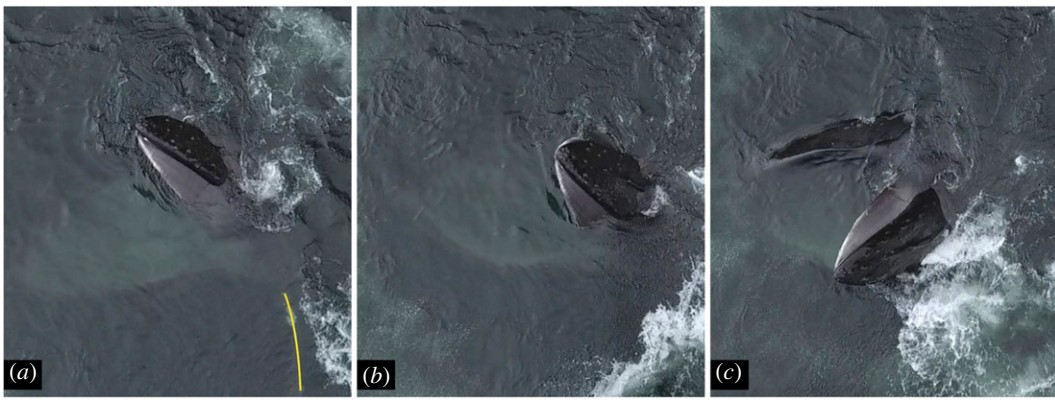

**Figure 4.** Photographic sequence of head tilt during the final portion of a lunge associated with horizontal pectoral herding by Whale A in Southeast Alaska. Movements progress from (*a*) earliest to (*c*) latest. Yellow line denotes the location of pectoral.

mouth, was $8 \pm 1$ s (calculated from 32 of 36 videos). Not all videos could be used to calculate lunge duration because they did not document the entire process.

We observed Whale A using horizontal pectoral herding in four locations that spanned approximately 21 km of coastline. This included Warm Springs Bay, Takatz Bay (2016 hatchery release site), Kasnyku Bay (2016 and 2017 hatchery release site) and Kelp Bay. In 2016, we observed Whale A lunge feeding in Warm Springs Bay, Takatz Bay and Kelp Bay. Although prey sampling was sparse and inconsistent, we observed juvenile salmon at all of these locations. In May 2016 and 2017, we collected juvenile hatchery salmon from Warm Springs Bay (within 12–44 days of feeding sessions) and visually identified juvenile salmon during all Warm Spring Bay foraging events. Feeding sessions in Takatz Bay coincided with a salmon release event and continued onto the day following. Juvenile salmon were only visually identified in Takatz Bay, but all feeding sessions were in the vicinity of hatchery salmon releases. In 2017, Whale A was observed horizontal pectoral herding in Kasnyku Bay and Kelp Bay. The feeding sessions in Kasnyku Bay were associated with salmon releases (within 7 days of a release). Prey sampling and otolith marks from fish collected within 1–3 days of feeding sessions confirmed juvenile hatchery salmon in the area. We collected juvenile salmon (hatchery and wild) within 8 days of feeding sessions in Kelp Bay. Pacific herring (*Clupea pallasii*) were also sampled in Kelp Bay during nine different feeding sessions. We were unable to differentiate whether prey being consumed in Kelp Bay were juvenile salmon or herring. Of all feeding sessions involving horizontal pectoral herding, 94.1% were identified as having targeted juvenile (hatchery-released chum and coho, wild pink (*Oncorhynchus gorbuscha*)) salmon.

## 3.2. Vertical pectoral herding

We documented Whale B (#2227 in Southeast Alaska Whale Catalog) solo bubble-net feeding at Hidden Falls Hatchery on 16 May 2017. During the 2.4 h observation period, we recorded 13 solo bubble-net feeding events, all of which were in the vicinity of newly released hatchery-reared juvenile coho salmon (figure 5). We observed two well-documented types of kinematic feeding behaviours for Whale B: vertical lunge and lateral lunge. We also documented vertical pectoral herding, which has not been previously documented in the scientific literature. Video footage depicting all three feeding types is provided in electronic supplementary material, S4.

Vertical pectoral herding was used in 30.8% of all feeding events. We identified vertical pectoral herding when the whale moved its pectorals from a neutral state (as in vertical lunge and lateral lunge) to a protraction–abduction posture (figure 6). After establishing this posture, the whale simultaneously moved both pectorals forward and into a V-shaped position on either side of its mouth, with pectorals curved ventrally (figure 7). A vertical lunge was used during 23.1% of all feeding events. When employing this technique, the whale's pectorals first abducted with the tips curved up. Prior to closing its mouth, the pectorals adducted to a vertical lunge position, tight against the side of the body. Finally, the pectorals retracted and angled posteriorly as the whale lunged to the surface (figure 7). The distinguishing feature between vertical lunge and pectoral herding was a slight upward dorsal-oriented curve to the pectorals and less visibility of the pectorals as they were abducted with a swept-back configuration. A lateral lunge was used in 46.2% of the feeding events

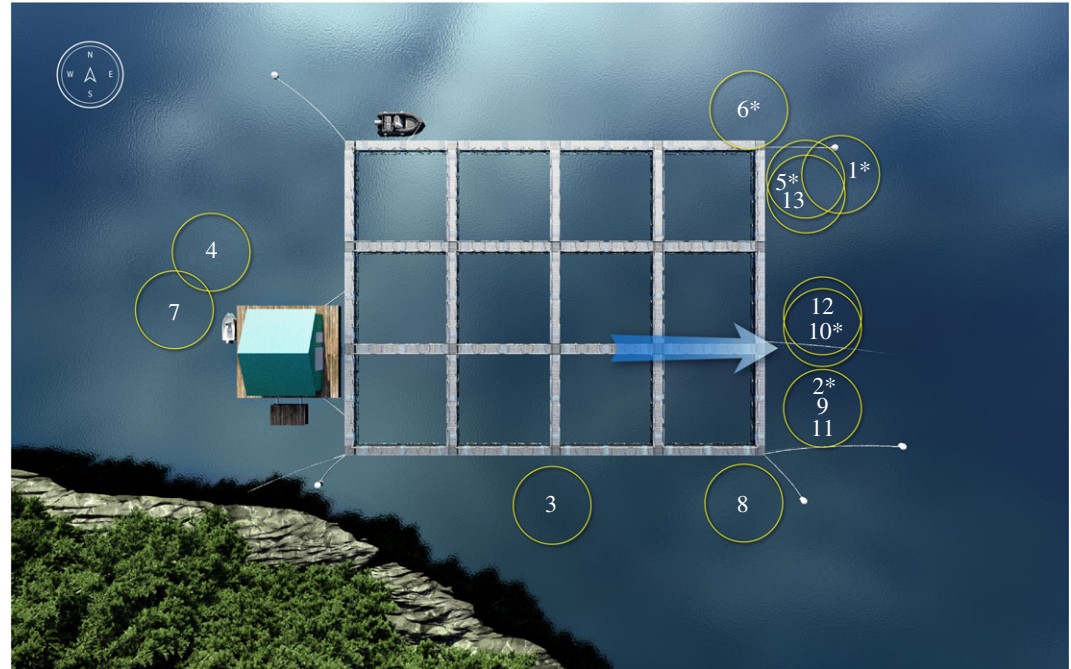

**Figure 5.** Graphical representation of the net pen structures at Hidden Falls Hatchery, in Kasnyku Bay (Southeast Alaska). Yellow circles represent bubble-nets created during feeding events for Whale B, numbered in chronological order. Blue arrow marks where juvenile coho salmon were being released into the marine environment. An asterisk denotes a feeding event conducted in sunlit waters. Events 1, 3, 4, 7, 11 and 12 involved a lateral lunge. Events 8, 9 and 13 involved a vertical lunge. Events 2, 5, 6 and 10 involved vertical pectoral herding. Graphic by Kyle Kosma.

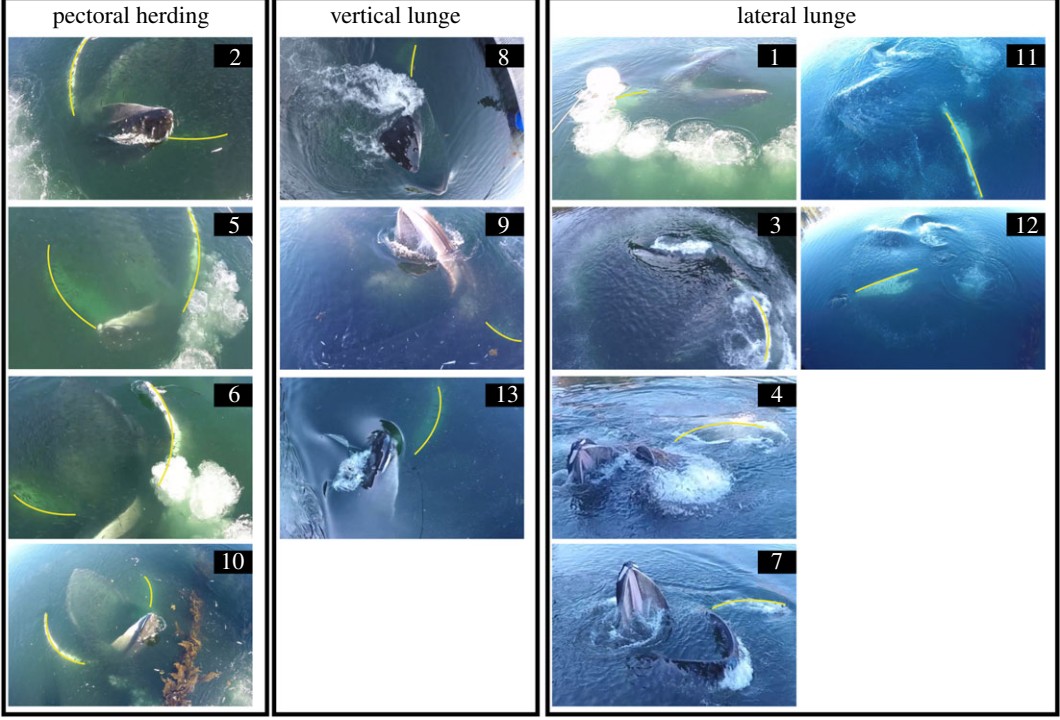

**Figure 6.** Snapshots from the footage of feeding events at Hidden Falls Hatchery, Kasnyku Bay (Southeast Alaska; 16 May 2017) by Whale B. Images are grouped according to three different kinematic feeding techniques at the conclusion of bubble-net formation: vertical pectoral herding, vertical lunge and lateral lunge. Events 2, 5, 6 and 10 involved vertical pectoral herding. Events 8, 9 and 13 involved a vertical lunge. Events 1, 3, 4, 7, 11 and 12 involved a lateral lunge. Yellow lines outline pectoral locations.

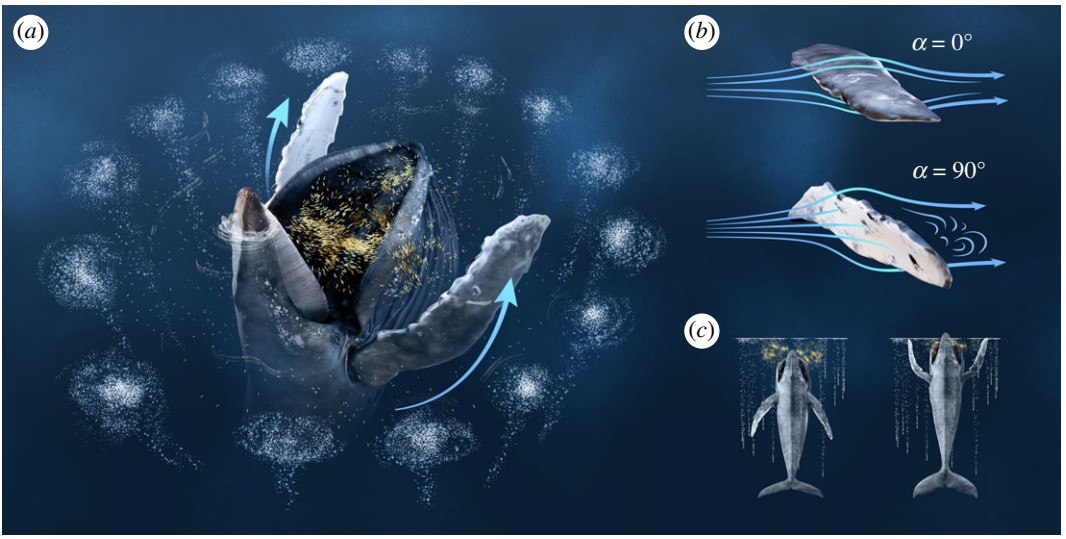

**Figure 7.** Graphical representation of vertical pectoral herding by Whale B in Southeast Alaska. Prey are denoted in yellow. (a) Whale deploys an upward-spiral bubble-net to corral prey and establish the first barrier; pectorals then protract to form a 'V' shape around the open mouth (depicted by blue arrows), creating a second physical barrier. (b) Change in the angle of attack ($\alpha$) from pre- (0°) to peri- (90°) vertical pectoral herding. (c) Body position comparison between pre- (left) and peri- (right) vertical pectoral herding. Graphic by Kyle Kosma.

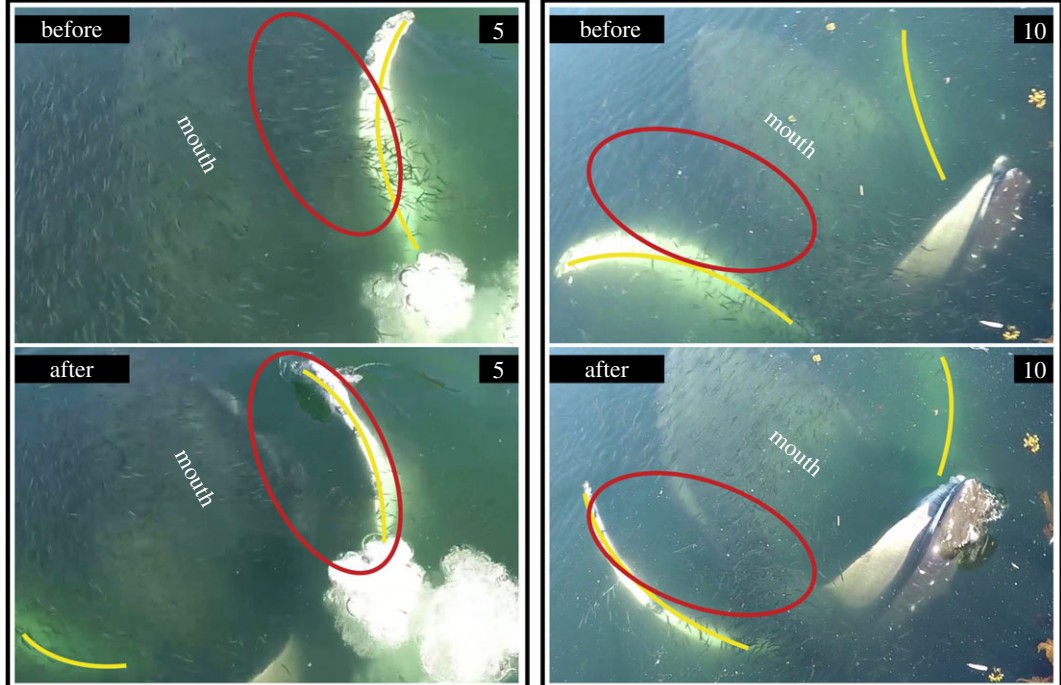

**Figure 8.** Before and after photographs of vertical pectoral herding by Whale B in Southeast Alaska (images relate to feeding events 5 and 10). Yellow lines denote pectorals. Red circles highlight the location of prey before pectoral movement and a gap in prey after pectoral movement.

(figure 6). When using this technique, the whale pivoted on its left pectoral and rolled approximately 90° while lunging. The left pectoral was exposed and occasionally broke the surface of the water as the whale used it to manoeuvre.

When documenting Whale B's feeding events, we observed notable differences in light conditions. Both vertical lunge (3 of 3) and lateral lunge (5 of 6) occurred in shaded waters. All vertical pectoral

herding events (4 of 4) occurred in sunlit water, which was easily identified from photographs due to a sun-induced green tint of the water (figure 6). Whale B employed different tactics in the same location only when light conditions varied. In general, Whale B appeared to use vertical pectoral herding in sunlit areas but switched to vertical lunge or lateral lunge when the same area became shaded. The single lateral lunge event in sunlight waters was located near a surface obstacle in the centre the bubble-net. Possible avoidance behaviour was documented as the whale lunged near the buoy. Prey movement in the direction opposite of vertical pectoral positioning was visible in 2 of 13 engulfment events (figure 8). In 'before' snapshots (i.e. images taken prior to vertical pectoral positioning), we observed a dense aggregation of prey between the mouth and pectoral. In 'after' snapshots (i.e. images taken once pectorals were placed in the V-shaped position), we observed less dense prey patches in the area between the mouth and pectoral. We also identified a greater relative density of prey that had moved towards the whale's mouth. We could not calculate lunge duration for Whale B because the whale started to lunge in water too deep to see the entire process using aerial footage. The variation in light conditions also prevented the identification of consistent cues for the start of a lunge.

# 4. Discussion

It is well known that humpback whale pectorals aid in acceleration and manoeuvrability during feeding events [27,28]. Our study recognizes an alternative use of pectorals during foraging. Here, we have provided the first empirical evidence for a longstanding hypothesis that humpback whales use their pectorals to herd and aggregate prey [38,43,50]. Our study combined the use of new technology and a unique viewing opportunity at Hidden Falls Hatchery to provide the vantage points necessary for such documentation. Although the concept that humpback whales use their pectorals to manipulate prey is not new, the use of pectorals in conjunction with a bubble-net (as a secondary barrier) had never been documented. Using direct video footage and photographic sequences, we described this foraging technique as 'pectoral herding', with two methods of execution: horizontal pectoral herding and vertical pectoral herding. We observed two humpback whales using bubble-nets as a primary barrier to corral prey, proceeded by deliberate movements of the pectorals to establish a secondary barrier before the lunge. These observations suggest that pectorals are used to further condense prey inside the bubble-net, thereby increasing feeding efficiency for each event. From our results, we found three ways in which humpback whales use pectorals to herd prey: (i) create a physical barrier to prevent evasion by prey, (ii) cause water motion to direct prey movement, and (iii) position the white coloration on the ventral side to reflect light, causing prey to move in the opposite direction [12,38]. These three methods of pectoral herding are not mutually exclusive and can be used in conjunction with one another.

## 4.1. Horizontal pectoral herding

The documented solo bubble-nets began and ended in the same general location. Thus, there is greater elapsed time for bubbles created near the beginning portion of the net, compared to the end. The greater dissipation of bubbles and possibility that fish are scared towards the beginning portion of the net (as a result of whale activity near the bubble-net closure site) suggests a potential weakness in the primary barrier. We hypothesize that Whale A uses horizontal pectoral herding to strengthen the beginning portion of the solo bubble-net and establish a secondary barrier to further condense prey, thereby increasing the amount of prey consumed during each lunge. Because the energetically costly movement of the left pectoral probably hinders the acceleration of the whale, we assert that an alternative use must be at play. We found that lunge durations of Whale A averaged 8 s, whereas Werth *et al.* [51] documented the mean engulfment rates from a solo humpback whale lunge to be closer to 2 s. This difference in engulfment rates with and without horizontal pectoral herding supports our hypothesis that any additional movement must substantially aid in prey capture. We conclude that Whale A used its pectorals in two of the three ways to herd prey: (i) create a physical barrier to prevent evasion by prey and (ii) cause water motion to direct prey movement. In addition, pectoral movements could create eddies and/or drag that increases the whale's capacity to alter prey movement. We note that our descriptions of horizontal pectoral herding rely upon observations from a single whale. However, we documented the use of this particular foraging technique by one additional whale, suggesting potential for cultural transmission of this foraging behaviour.

In over half of the documented events, Whale A rotated its head in the direction of the left pectoral before closing its mouth (during all other fully documented events, the head remained centred and never rotated in the opposite direction). This suggests that the left pectoral was herding prey and that the whale turned its mouth into the path of swimming prey, further increasing the amount of fish consumed per lunge. The lower jaw turned at an angle that exposed prey to the largest circumference of the buccal cavity, which probably prevented escape between the lower jaw and the surface of the water. The rostrum was also above the surface of the water to avoid blocking prey from entering the buccal cavity when the whale turned its head. When the whale's head remained central, the lower jaw surfaced to meet the upper jaw. During these events, the whale may have sensed that its buccal cavity was full of fish, making head rotation counterproductive [52].

## 4.2. Vertical pectoral herding

Our current understanding about lunge feeding revolves around the theory that whales use their pectorals to actively increase lift and/or stabilize their body during a lunge. The pectoral position used by Whale B suggests that the whale violated two out of the four criteria proposed for a hydrodynamic stroke [40]. First, the pectorals were not oriented at an efficient angle into the path of the stroke ($\alpha > 17.5°$). The stroke was also not oriented perpendicular to the body, which would inhibit stability during the lunge. Therefore, we claim that the pectoral movements of Whale B were not intended to increase hydrodynamic efficiency, stability or lift. Whale B's forward speed was probably hindered by a high angle of attack and V-shaped position of the pectorals around the mouth. During three of the four pectoral herding events, the rostrum and left pectoral broke the surface of the water at approximately the same time (within 1 s of each other). There is no hydrodynamic reason for the pectorals to be in line with or above the position of the mouth during a lunge. By eliminating the use of pectorals for stabilization and thrust, we deduced that Whale B's pectorals were used to create a secondary barrier along the edges of the mouth during a lunge, manipulating prey movement towards the mouth and increasing foraging efficiency.

Light conditions and prey reactions also suggest that Whale B used its pectorals to herd prey. There were three main locations around the net pens that had recurring feeding events. During Whale B's feeding session, the eastern side of the net pens transitioned from sunlit waters to shade. In all three of these locations, Whale B used vertical pectoral herding when lunging in the sun. During the only sunlight feeding event without vertical pectoral herding, we hypothesize that Whale B was manoeuvring around a buoy and that the whale would have used vertical pectoral herding if the obstacle were not present. When waters transitioned from sunlight to shade in these three main locations, the whale used vertical or lateral lunges instead of vertical pectoral herding. This provides support for the hypothesis that behavioural shifts were based on light conditions rather than locational differences. Brodie [38] suggested that the ventral side of the pectorals can be used to 'flash' fish and cause them to move in the direction of the dark mouth, which functions as a deceptive refuge. When prey movement was visible in sunlit waters, we observed prey moving in the direction of the mouth, apparently in response to the position of the pectorals. This is convincing evidence that pectorals alter prey behaviour. The lack of vertical pectoral herding in shaded water suggests that the physical presence of the pectorals alone is not effective enough to cause fish to move towards the mouth. The combination of light reflection and a physical barrier probably provides a foraging benefit to justify the hydrodynamic detriment caused by vertical pectoral herding. Thus, it is probable that Whale B used pectorals in two of the three ways to herd prey: (i) create a physical barrier to prevent evasion by prey and (ii) position the white coloration on the ventral side to reflect light and cause prey to move in the opposite direction [12,38].

## 4.3. Prey and behavioural plasticity

Schooling fish cluster in response to predators or other startling disturbances [53–57], and humpback whales have been known to take advantage of this behaviour [26]. Sharpe [15] experimented with an artificial pectoral and found that herring respond to a rotating pectoral by fleeing in the opposite direction. It has also been suggested that humpback whales manipulate prey by slapping their pectoral fins or flukes on the surface of the water [7,26]. Whale A's pectoral movement makes a startling disturbance that could alter the direction of prey within the bubble-net barrier. We were unable to see prey in videos of Whale A foraging. However, the continued use of horizontal pectoral herding, in combination with its hydrodynamic disadvantages, is strong evidence for an increase in

foraging efficiency. Additionally, a study on hatchery-reared juvenile salmon [58] showed that fish avoid light and seek out dark refugia when artificial lights were activated and/or flashing. We believe that light reflected off the ventral surface of Whale B's pectorals served as a stimulus to scare fish in the direction of the dark 'refuge' of the whale's mouth. We were able to directly observe prey movement towards the mouth in response to Whale B's pectoral placement in some of the videos. Pectoral movement and flashing may directly stun or disorient prey [7].

It is well known that humpback whales use bubble-nets to aggregate prey [12,26]; however, bubble-nets may not be as efficient when prey do not naturally aggregate into dense patches. This is because schooling fish would aggregate within a single area of the bubble-net, enabling the consumption of most fish in a single lunge. Non-schooling fish may very well distribute themselves throughout the bubble-net, resulting in fewer fish consumed per lunge. Acoustic prey surveys at our study site showed that groups of juvenile coho (*Oncorhynchus kisutch*) and chum (*Oncorhynchus keta*) salmon were small, patchy and short-lived compared to those formed by herring and krill [59]. Whales tend to moderate their behaviour to efficiently exploit different prey types and respond to dynamic prey conditions [14,60]. It is possible that the two whales we observed have independently altered their foraging strategies to accommodate non-schooling fish and more effectively incorporate hatchery-released juvenile salmon into their diets. Because aerial documentation of solo bubble-netting whales has been limited, we cannot conclude whether or not pectoral herding is restricted to these whales and the unique prey resource of hatchery-reared juvenile salmon. Pectorals are an efficient secondary barrier and may be used by other whales lunging on different prey. For Whales A and B, 93.9% of pectoral herding events exclusively targeted juvenile salmon. The remaining events may have also targeted herring as prey. Additionally, a bubble-net may be substantially larger than the size of a whale's open mouth, restricting engulfment to only a portion of the prey enclosed within the net. A secondary barrier further condenses prey, conceivably enhancing the energy gained per lunge.

McMillan *et al.* [18] documented humpback whales using a feeding strategy called 'trap-feeding'. The authors inferred that whales use pectorals to manipulate prey by flicking fish into their mouth. The available footage of the pectoral movement in this study relies on a lateral perspective with poor visibility below the water's surface and no view of prey. This makes it difficult to connect pectoral movements to a specific behaviour or make inferences about prey responses. Additionally, lateral footage makes it difficult to differentiate between the use of pectorals as a stabilizing force during a lunge and pectoral movements to manipulate prey. In general, most whale observations are obtained from land or boat, yielding lateral views that limit the perspective and skew our perception of individual behaviours. With innovative technology (e.g. UAVs, small video cameras), we can now gain the perspectives necessary for more accurate interpretations of marine mammal foraging tactics. Our observations, which relied on an aerial perspective, provide insight into the position of humpback whales in relation to prey (above and below the water) as well as a more detailed depiction of the whale's movements and position during feeding events. Based on lateral–aerial comparisons of pectoral herding by humpback whales, we believe that conventional boat or land-based footage should be supplemented by aerial imagery in order to gain insight and avoid misinterpretations about marine mammal behaviour.

Despite the advantages of using advanced technology, our study is limited by small sample sizes and a lack of quantitative kinematics. Our findings depended on functional interpretations of movements made by two whales with only above-surface documentation. A more inclusive survey of solo feeding humpback whales (encompassing broader spatial scales and additional whales) would provide greater insight into how these animals are taking advantage of their lengthy appendages during foraging. Furthermore, future investigations should pair aerial footage of feeding whales with prey distribution data, and synchronous motion suction cup tags (i.e. DTAGs) to better quantify kinematic behaviours and prey dynamics, both above and below the surface [61]. Notably, however, our study suggests a flaw with current tagging technology. Although tags are often deployed on the backs of whales to record movements (pitch, yaw and roll) of the entire whale, we found that prey aggregation and capture is not limited to movements of the head, caudal peduncle and tail flukes. Thus, tag sensors that also quantitatively record these movements of the pectorals would allow for a clearer understanding of how these appendages are kinematically being used. Finally, more accurate lunge durations (e.g. starting when the whale's mouth opened) would help us compare acceleration rates between lunges with and without pectoral herding, furthering our understanding about the hydrodynamic impacts caused by pectoral movements.

In summary, our results provide empirical evidence of the use of pectorals to herd prey. They also illustrate considerable variation among individual humpback whale foraging strategies. With our

documentation of pectoral herding, we have provided support for plasticity in foraging behaviour of cetaceans. These animals are highly innovative, with individual whales successfully using different tactics to approach the same prey in the same situation [26]. Maintaining a suite of foraging strategies probably aids humpback whales in a changing environment, where food availability fluctuates and competition may impact population dynamics. Further investigation would enhance our understanding about whether humpback whales use pectoral herding as a principal foraging technique as well as the conditions that promote its use.

Ethics. This research was conducted under National Marine Fisheries Service (NMFS) permits 14122 and 18529, University of Alaska Institutional Animal Care and Use Committee (IACUC) permit 907314-3 and State of Alaska Department of Fish and Game permit CF-18-049. The research reported in this publication was supported by the National Institute of General Medical Sciences of the National Institutes of Health, under award RL5GM118990. The content within this manuscript is solely the responsibility of the authors and does not necessarily represent the official views of the National Institutes of Health.

Data accessibility. All data and videos for this paper are attached via electronic supplemental material.

Authors' contributions. M.M.K., A.J.W., A.R.S., and J.M.S. contributed to the experimental conception and design; M.M.K. was primarily responsible for data collection; M.M.K was primarily responsible for analysing and interpreting data with contributions from A.J.W., A.R.S. and J.M.S.; M.M.K. drafted the article. All authors revised the article and provided final approval for the version to be published.

Competing interests. We declare we have no competing interests.

Funding. M.M.K. was supported by the Biomedical Learning and Student Training (BLaST) programme through the University of Alaska Fairbanks. Fieldwork was funded by the Alaska Whale Foundation (AWF), Lindblad Expeditions–National Geographic Conservation Fund and Mark Kelley Photography. Research reported in this publication was supported by the National Institute of General Medical Sciences of the National Institutes of Health, under award no. RL5GM118990. The content within this manuscript is solely the responsibility of the authors and does not necessarily represent the official views of the National Institutes of Health.

Acknowledgements. Leonie Mahke, Nigel Ogle, Jasmine Gil, Christine Walder, Danielle Derrick and Cheryl Barnes provided field support. Northern Southeast Regional Aquaculture Association and Hidden Falls Hatchery staff provided logistical support and net pen access necessary for obtaining aerial footage. Kyle Kosma (kylekosma.com) helped analyse feeding methods to create graphical representations (figures 2, 5 and 7). Additional support was provided by Jennifer Cedarleaf, Ellen Chenoweth and Lauren Wild (The Human Cetacean Interaction Lab at the University of Alaska Southeast). Cheryl Barnes, Megan McPhee and Matthew Wooller reviewed earlier drafts of this manuscript.

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
