## [Reviewer comments · Royal Society Open Science]

Review History

RSOS-191104.R0 (Original submission)

Review form: Reviewer 1

Is the manuscript scientifically sound in its present form?

Yes

Are the interpretations and conclusions justified by the results?

Yes

Is the language acceptable?

Yes

Do you have any ethical concerns with this paper?

No

Have you any concerns about statistical analyses in this paper?

No

Recommendation?

Accept with minor revision (please list in comments)

Comments to the Author(s)

Please see attached (Appendix A).

Review form: Reviewer 2

Is the manuscript scientifically sound in its present form?

No

Are the interpretations and conclusions justified by the results?

No

Is the language acceptable?

Yes

Do you have any ethical concerns with this paper?

No

Have you any concerns about statistical analyses in this paper?

No

Recommendation?

Reject

Comments to the Author(s)

The authors describe humpback whale behavior at a salmon hatchery with a focus on flipper movements. Humpback flippers are interesting structures that can produce significant hydrodynamic forces for a wide range of functions from social signaling to effecting both foraging and non-foraging maneuvers. The manuscript is well written but the data is very hard to understand and interpret as it is currently displayed and described in the manuscript. Some of the methods described for taking photo/video of whale behavior is incomplete, such as filming frequency, and the photos shown in figures 3, 4, 6, 8 do not show time stamps or time intervals. No video is provided for review. The schematics added in figure 2 and figure 7 do not add much given the lack of actual data presentation. Time scale of flipper movements should at least be a minimum for description of behavior given the focus of the manuscript. If drone data had RTK that should be used in coordination with altimeter data for kinematic analysis, if possible.

The title should reflect that the observed behavior is not natural. This study is an interesting experiment of how humpback whales exploit an artificial food source, but it is unclear how broadly relevant the present results are to our understanding of how whales use flippers in natural contexts on natural prey and in natural distributions/abundances.

The first paragraph of the discussion describes new technology used in this study. What technology are the authors referring to exactly? The last paragraph on page 8 states that tag deployed on whale dorsal surfaces only record movements of whole whale. There are papers recently published I suggest the authors refer to going back to 2017 and 2016 that describe the use of cameras within tags that show movements of jaws and appendages relative to the body.

Decision letter (RSOS-191104.R0)

28-Aug-2019

Dear Ms Kosma

On behalf of the Editors, I am pleased to inform you that your Manuscript RSOS-191104 entitled "Pectoral herding: an innovative tactic for humpback whale foraging" has been accepted for publication in Royal Society Open Science subject to minor revision in accordance with the referee suggestions. Please find the referees' comments at the end of this email.

The reviewers and handling editors have recommended publication, but also suggest some minor revisions to your manuscript. Therefore, I invite you to respond to the comments and revise your manuscript.

- Ethics statement

- Data accessibility

If you wish to submit your supporting data or code to Dryad (<http://datadryad.org/>), or modify your current submission to dryad, please use the following link:
<http://datadryad.org/submit?journalID=RSOS&manu=RSOS-191104>

- Competing interests

- Authors' contributions

- Acknowledgements

- Funding statement

Because the schedule for publication is very tight, it is a condition of publication that you submit the revised version of your manuscript before 06-Sep-2019. Please note that the revision deadline will expire at 00.00am on this date. If you do not think you will be able to meet this date please let me know immediately.

- 1) A text file of the manuscript (tex, txt, rtf, docx or doc), references, tables (including captions) and figure captions. Do not upload a PDF as your "Main Document";
- 2) A separate electronic file of each figure (EPS or print-quality PDF preferred (either format should be produced directly from original creation package), or original software format);
- 3) Included a 100 word media summary of your paper when requested at submission. Please ensure you have entered correct contact details (email, institution and telephone) in your user account;
- 4) Included the raw data to support the claims made in your paper. You can either include your data as electronic supplementary material or upload to a repository and include the relevant doi

within your manuscript. Make sure it is clear in your data accessibility statement how the data can be accessed;

5) All supplementary materials accompanying an accepted article will be treated as in their final form. Note that the Royal Society will neither edit nor typeset supplementary material and it will be hosted as provided. Please ensure that the supplementary material includes the paper details where possible (authors, article title, journal name).

on behalf of Dr Denise Greig (Associate Editor) and Kevin Padian (Subject Editor)
openscience@royalsociety.org

Associate Editor Comments to Author (Dr Denise Greig):

Associate Editor: 1

Comments to the Author:

Thank you for this interesting submission. I enjoyed the graphics and the video. I agree with suggesting from both reviewers that you may want to describe the speed of some of the movements documented in the figures. I just had two minor comments on the text:

Page 5, line 8...I would say "newly documented" rather than "novel" because we do not know if it is new to the whales.

Page 5, line 55...I would split the sentence starting with "In 2016, we observed..." into two sentences or add a colon.

Reviewer comments to Author:

Reviewer: 1

Comments to the Author(s)

Please see attached

Reviewer: 2

Comments to the Author(s)

The authors describe humpback whale behavior at a salmon hatchery with a focus on flipper movements. Humpback flippers are interesting structures that can produce significant hydrodynamic forces for a wide range of functions from social signaling to effecting both foraging and non-foraging maneuvers. The manuscript is well written but the data is very hard to understand and interpret as it is currently displayed and described in the manuscript. Some of the methods described for taking photo/video of whale behavior is incomplete, such as filming frequency, and the photos shown in figures 3, 4, 6, 8 do not show time stamps or time intervals. No video is provided for review. The schematics added in figure 2 and figure 7 do not add much given the lack of actual data presentation. Time scale of flipper movements should at least be a minimum for description of behavior given the focus of the manuscript. If drone data had RTK that should be used in coordination with altimeter data for kinematic analysis, if possible.

The title should reflect that the observed behavior is not natural. This study is an interesting experiment of how humpback whales exploit an artificial food source, but it is unclear how broadly relevant the present results are to our understanding of how whales use flippers in natural contexts on natural prey and in natural distributions/abundances.

The first paragraph of the discussion describes new technology used in this study. What technology are the authors referring to exactly? The last paragraph on page 8 states that tag deployed on whale dorsal surfaces only record movements of whole whale. There are papers recently published I suggest the authors refer to going back to 2017 and 2016 that describe the use of cameras within tags that show movements of jaws and appendages relative to the body.

Author's Response to Decision Letter for (RSOS-191104.R0)

See Appendix B.

Decision letter (RSOS-191104.R1)

23-Sep-2019

Dear Ms Kosma,

I am pleased to inform you that your manuscript entitled "Pectoral herding: an innovative tactic for humpback whale foraging" is now accepted for publication in Royal Society Open Science.

You can expect to receive a proof of your article in the near future. Please contact the editorial office (openscience_proofs@royalsociety.org and openscience@royalsociety.org) to let us know if

you are likely to be away from e-mail contact -- if you are going to be away, please nominate a co-author (if available) to manage the proofing process, and ensure they are copied into your email to the journal.

on behalf of Dr Denise Greig (Associate Editor) and Kevin Padian (Subject Editor)
openscience@royalsociety.org

Appendix A

Review of *Pectoral herding: an innovative tactic for humpback whale foraging*

This study provides the first direct evidence for the ‘pectoral herding’ behavior of humpback whales. Bubble netting humpbacks wave their flippers in a non-hydrodynamically efficient way to scare schools of juvenile fish into their open mouths. Although this behavior has been previously hypothesized, it has not been otherwise confirmed. This study demonstrates video evidence of the flipper motion, how the prey responds to the flipper motion, and how it differs from other documented flipper uses. This paper is well written and very comprehensive. The authors sort through the different hypotheses for humpback flipper movements and convincingly demonstrate evidence for pectoral herding. Furthermore, the authors are careful to acknowledge the weaknesses of this study (n=2 individuals). I recommend this study for publication in RSOS and have only minor comments.

Observations:

These appear to be pretty slow lunges. Likewise, the flipper movements of whale B seem pretty slow compared to the thrust-generating flaps found in [25]. Particularly in your discussion pertaining to vertical herding I think the speed of the lunges and the flippers further supports your point of this not being a hydrodynamically active stroke. Since you’re not measuring the speeds you can only say so much, but it may be worth mentioning the slower speeds involved.

The complex flipper movement of whale A is really interesting, and so is the head tilt towards the moving flipper. Furthermore, this whale is practically not moving forward at all. It would be really interesting to know what the back flipper is doing.

Please do ensure that the important videos are attached in the supplementary material or a repository (currently they are available on dropbox), since they are very important to this paper. Not all the videos need to be presented, just the ones that you show in the figures.

Minor comments:

The introduction could use some additional copy editing. A few examples: 1) a comma after ‘sizable’, 2) the “;” after “prey with lunge feeding” should probably be a “:”. There are a few more instances as well. In this respect, the rest of the paper is pretty good.

Pg7 “Our current understanding about lunge feeding revolves around the theory that whales *must either* actively increase lift with their pectorals and/or use them to stabilize their body during a lunge.” I think this sentence is a bit misleading. During lunge feeding, it has been shown that humpback flippers can be used to generate additional thrust and it has been hypothesized (although it is almost certain) that they can be used for stability and maneuvering. Additionally, they could also play a hydrodynamically passive role during some lunging events and either be held in a neutral position or their motion could be a reaction to the accelerative and decelerative forces of the body. I think ‘*must either*’ is too simplistic and I would rephrase the sentence.

Pg8 “It is well known that humpback whales aggregate their prey with a bubble-net.” I think this paragraph is confusing. Has it actually been shown that “bubble-nets are rendered more inefficient if the prey does not naturally aggregate into dense patches”? Isn’t this the point of the bubble netting? I get the gist, but I would rephrase the first few sentences of the paragraph.

Pg8: “However, our study reveals a flaw with the current tagging technology. Our findings illustrate that prey aggregation and capture are not limited to movements of the head, caudal peduncle, and tail flukes, but current tagging protocol deploys the tag on the back of the whale, recording only movements of the whole whale.” A slight caution with this sentence: camera tags do allow for examination of the flippers and have been around for several years (see *The role of flippers, flukes, and body flexibility in blue whale maneuvering performance*).

Figure 4 would benefit from highlighting the left flipper to show that the head roll is in the motion of the flipper.

Appendix B

Pectoral herding: an innovative tactic for humpback whale foraging

Madison M. Kosma¹, Alexander J. Werth², Andrew R. Szabo³, Janice M. Straley⁴

¹College of Fisheries and Ocean Sciences, University of Alaska Fairbanks, Juneau, AK 99801

²Department of Biology, Hampden-Sydney College, Hampden-Sydney, VA 23943

³Alaska Whale Foundation, Petersburg, AK 99833

⁴Department of Natural Sciences, University of Alaska Southeast, Sitka, AK 99835

Abstract

Humpback whales (*Megaptera novaeangliae*) have exceptionally long pectorals (*i.e.*, flippers) that aid in shallow water navigation, rapid acceleration, and increased manoeuvrability. The use of pectorals to herd or manipulate prey has been hypothesized since the 1930s. We combined new technology and a unique viewing platform to document the additional use of pectorals to aggregate prey during foraging events. Here, we provide a description of “pectoral herding” and explore the conditions that may promote this innovative foraging behaviour. Specifically, we analysed aerial videos and photographic sequences to assess the function of pectorals during feeding events near salmon hatchery release sites in Southeast Alaska (2016 to 2018). We observed the use of solo bubble-nets to initially corral prey, followed by calculated movements to establish a secondary boundary with the pectorals – further condensing prey and increasing foraging efficiency. We found three ways in which humpback whales use pectorals to herd prey: 1) create a physical barrier to prevent evasion, 2) cause water motion to guide prey toward the mouth, and 3) position the ventral side to reflect light and alter prey movement. Our findings suggest that behavioural plasticity may aid foraging in changing environments and shifts in prey availability. Further study would clarify if “pectoral herding” is used as a principal foraging tool by the broader humpback whale population and the conditions that promote its use.

Background

The metabolic demand of a large body size forces baleen whales to require sizable dense prey patches. Large body sizes of baleen whales generate high metabolic demands that require the consumption of sizable, dense patches of prey [1-3]. However, filter feeding on these patches is energetically demanding for these animals and requires inventive effective methods for prey aggregation [2]. In forquals, behavioural plasticity and foraging innovations are common among rorquals [4,5]. H-and humpback whales (*Megaptera novaeangliae*) provide an excellent example of how individual advances-changes in behaviour can lead to diverse foraging tactics that maximize feeding efficiency [6- 9]. They are known for a wide variety of foraging strategies including: Such foraging includes lunge feeding [6,10], bubble-net feeding [6,11-14], flick feeding [6], cooperative feeding [15], lobtail feeding [7] and other idiosyncratic tactics [12,16-18].

Humpback whales are one of the world’s largest filter-feeders and regularly use lunge feeding to capture prey with lunge feeding; an-This particular technique is energetically costly [19] and sequential two-step process requires a two-step process. The whale first uses a high-velocity lunge to engulf large volumes of prey-laden water. The whale then closes its mouth and the baleen acts as a sieve to filter prey [14,20]. First, during a high-velocity lunge the whale engulfs a large volume of prey-laden water and second the prey is filtered from the water with specialized feeding anatomy [14,20].-The lunge can occur at depth [2,10,20-22] or on the surface [7,23,24]. In-and in both locations-situations lunge feeding requires acceleration to high speeds [2,26] because the animal must overcome considerable drag from an open mouth-[20,22]. To counteract drag and increase speed, they-humpback whales open their mouths gradually, in synchrony with strong fluke strokes

[20,22]. ~~This~~ acceleration maximizes the amount of water engulfed and aids in the capture of active prey [26]. Humpback whales feeding near the surface exhibit an array of lunge types [6,12,15] and some are in association with the creation of bubbles. A bubble-net is denoted by the formation of a ring of bubbles in a clockwise fashion to enclose prey [6,7,12,13,27] and this strategy can be employed by an individual or a group of whales. Bubble nets serve as a physical barrier to increase lunge efficiencies and are most commonly used on naturally schooling fish (i.e., Pacific herring). ~~This bubble barrier makes the whale's lunge more efficient and is most commonly used on naturally schooling fish (i.e., Pacific herring).~~

Humpback whales have a distinctive body morphology that allows for the ~~successful and~~ efficient capture of prey [28,29]. Notably, they have the longest pectorals (i.e., flippers) of any cetacean, measuring from one-quarter to one-third of their body length [30,31]. ~~The pectorals of other cetaceans typically, whereas other cetaceans' pectorals do not exceed one-seventh the length of their bodies [32]. These exceptionally long appendages of humpback whales allow for effective navigation in shallower waters [32,33], quick-rapid acceleration, and increased greater manoeuvrability and increased stability [6,34,35], thereby increasing capture abilities of small prey such as-~~ Each component enhancing their ability to capture prey such as euphausiids, herring (*Clupea* spp.), capelin (*Mallotus villosus*), and sand lance (*Ammodytes* spp.) [36,32,37,38]. ~~However, larger pectorals do have a hydrodynamic disadvantage and if not used effectively during foraging they can be detrimental to prey capture- f not positioned effectively, however, larger pectorals may present a hydrodynamic disadvantage by increasing drag[39].~~

As the buccal cavity expands during a lunge, a hydrodynamically optimal position for the pectorals is for one or both to extend with the leading edge held at low angles of attack (α) [40]. Positioning the pectorals in this manner minimizes drag and provides the greatest amount of lift. The perpendicular position of extended pectorals also stabilizes the whale's body during a lunge [40]. During a roqual lunge, the optimal pectoral position is one or both pectorals extended during ventral pouch expansion and the pectorals' leading edges held at low angles of attack ($\alpha \sim 0^\circ$) [40]. Positioning the pectorals in this manner is hydrodynamically efficient, corresponding to the greatest amount of lift and least amount of drag. The extended perpendicular position of the pectorals can also stabilize the body during a lunge [40]. Additionally, it has been hypothesized that rapid pectoral movement just prior to a lunge generates an upward pitching motion that counteracts the torque caused by rapidly engulfing water [35,40]. Segre *et al.* [25] defined four conditions for pectoral movement that would generate lift and increase propulsive thrust during an engulfment event: 1) both pectorals must move symmetrically, 2) pectorals are angled into the path of the stroke, 3) the stroke is oriented perpendicular to the whale's body, and 4) the stroke is aligned with the direction of travel [25]. Lift is generated as pectorals are rotated at an angle to the water flow (angle of attack or α). However, this angle must be small relative to the direction of travel [41]. Above a critical α , the pectoral will impede lift, making the movement detrimental to acceleration. Miklosovic [42] found that peak hydrodynamic efficiency of a humpback whale pectoral is around $\alpha = 7.5^\circ$. Above this, drag increases and lift decreases, with complete stall occurring at $\alpha \sim 17.5^\circ$. For symmetrical pectorals, lift is created as one pectoral is canted at an angle to the water flow (angle of attack or α), however, lift is only generated in an efficient manner when the pectorals are oriented at a small angle into their path of travel [41]; if the pectoral is canted at too high of an α , drag will increase. For a whale's pectoral to be an effective biological hydrofoil, it must produce substantial lift while minimizing drag [31]. Above a critical α , the pectoral will impede lift to a point of stall, making the movement detrimental to acceleration. Miklosovic [42] illustrated that the peak hydrodynamic efficiency of a humpback whale pectoral is around $\alpha \sim 7.5^\circ$ and above this drag will increase and lift will decrease (with complete stall occurring at $\alpha \sim 17.5^\circ$). These studies illustrate that there are strict hydrodynamic criteria for using pectorals efficiently during lunge-feeding.

Formatted: Font: Italic

In addition to providing lift, decreasing drag, and promoting acceleration, pectorals may be used to corral or concentrate prey during lunge-feeding events. Humpback whales have multiple foraging strategies to aggregate prey, but concentration of prey may be increased by herding techniques [32,43]. Howell [44] was the first to suggest that humpback whales use their pectorals to direct schools of fish into their mouths. Brodie [39] elaborated on this theory by describing the use of white coloration on the pectoral's ventral surface to "flash" fish and herd prey toward the whale's mouth. He stated, "if there are hydrodynamic disadvantages to such large flippers there must be selective compensation, one possibility being their role in concentrating prey" [39]. Both authors, however, reported reservations about their findings because they lacked the perspective necessary to document such behaviours [39]. Our objective was to use new technology (e.g., unoccupied aerial vehicles, small mobile-video cameras) to document and describe the distinctive role of humpback whale pectorals in herding and aggregating prey. We focused our efforts on whales feeding near salmon hatchery release sites [45] in Southeast Alaska (2016 to 2018). Hatchery structures allowed for close approaches with minimal behavioural disruption. Our results enhance our understanding of the complex and innovative foraging tactics that may be critical to humpback whale survival as population dynamics and environmental conditions continue to change [46,47].

Methods

Study Location and Timing

This study was conducted in Chatham Strait, along the eastern shore of Baranof Island in Southeast Alaska (Fig. 1). We conducted systematic surveys from Warm Springs Bay north to Kelp Bay, with an emphasis on salmon hatchery release sites in Takatz Bay and Kasnyku Bay in 2016 (mid-May through the end of June) and 2017 (mid-April through end of July). We put forth a more directed effort to document foraging strategies by humpback whales in Kasnyku Bay in 2018 (May). All effort was timed to overlap with releases of juvenile salmon from Hidden Falls Hatchery (managed by the Northern Southeast Regional Aquaculture Association).

Data collection

We collected humpback whale sightings and behavioural observations as part of a three-year study (2016 to 2018) of humpback whale predation at Hidden Falls Hatchery and surrounding areas. We took identification photographs of each whales using digital SLR cameras with lenses ranging in focal lengths from 70mm to 300mm. Humpback whales were individually identified based on the pigmentation and trailing edges of their flukes and/or the shape and marks of their dorsal fins [48] and cross-referenced with the Southeast Alaska Humpback Whale Catalog [49]. This catalogue included all whale sightings through 2012 and additional observations from later time periods (Straley & Gabriele unpubl.). We made an effort to capture video and photographic sequences with a Nikon D7000 camera whenever whales were observed feeding at the surface. In 2017, we also used a GoPro Hero5 Black video camera affixed to the end of a 3.5-meter pole to provide an aerial perspective while standing on walkway platforms attached to hatchery net pens. These platforms provided a unique and close-up perspective without disturbing whale behaviour that enabled camera views directly above or within bubble-nets created by the feeding whales. In 2018, we used an unoccupied aerial vehicle (UAV; DJI Mavic Pro with 4k video at 24fps) to capture footage of whales surface lunge feeding near the facility. In addition to visual prey identification, we used a cast net and herring jig to sample prey in foraging areas. We removed juvenile salmon otoliths to differentiate hatchery-reared and wild origin fish according to methods described by Volk *et al.* [50].

Data Analysis

Commented [MK1]: We combined the last two paragraphs to make the background section more concise and easier to read. Additionally, we took out "novel" as the editor requested and added the third potential use of pectorals as hydrodynamically passive (or decreasing drag with there placement) requested by reviewer 1

[revised manuscript text omitted]

427
428
429

Commented [MK2]: Edited to make it clearer what is missing from tagging technology (suggestion from both reviewers).

[revised manuscript text omitted]

560 A. Ward), pp. 193-204. Boston: Dr W. Junk Publishers.

57. Pitcher TJ, Parrish JK. 1993 Functions of shoaling behaviour in teleosts. Pages 363–439 in T. J. Pitcher,
ed. Behaviour of teleost fishes, 2nd ed. Chapman & Hall, New York.
58. Nemeth RS, Anderson JJ. 1992 Response of Juvenile Coho and Chinook Salmon to Strobe and Mercury
Vapor Lights. *North Am. J. Fish. Manag.* **12**, 684–692. (doi: 10.1577/1548-
8675(1992)012<0684:ROJCAC>2.3.CO;2)
59. Chenoweth EM. 2018 Bioenergetic and Economic Impacts of Humpback Whale Depredation at Salmon
Hatchery Release Sites [dissertation]. Fairbanks (AK): University of Alaska Fairbanks.
60. Cade DE, Friedlaender AS, Calambokidis J, Goldbogen JA. 2016 Kinematic Diversity in Rorqual Whale
Feeding Mechanisms. *Curr. Biol.* **26**, 2617–2624. (doi:10.1016/j.cub.2016.07.037)
61. Johnson MP, Tyack PL. 2003 A digital acoustic recording tag for measuring the response of wild marine
mammals to sound. *IEEE J. Ocean. Eng.* **28**, 3– 12. (doi:10.1109/JOE.2002.808212)

572